# High Drug Resistance Levels Compromise the Control of HIV Infection in Pediatric and Adult Populations in Bata, Equatorial Guinea

**DOI:** 10.3390/v15010027

**Published:** 2022-12-21

**Authors:** Ana Rodríguez-Galet, Judit Ventosa-Cubillo, Verónica Bendomo, Manuel Eyene, Teresa Mikue-Owono, Jesús Nzang, Policarpo Ncogo, José María Gonzalez-Alba, Agustín Benito, África Holguín

**Affiliations:** 1HIV-1 Molecular Epidemiology Laboratory, Microbiology and Parasitology Department, Hospital Ramón y Cajal-IRYCIS and CIBEREsp-RITIP-CoRISpe, 20834 Madrid, Spain; 2Fundación Estatal, Salud, Infancia y Bienestar Social (CSAI), 28029 Madrid, Spain; 3Unidad de Referencia de Enfermedades Infecciosas (UREI), Hospital Regional de Bata, Bata 88240, Equatorial Guinea; 4Laboratorio de Análisis Clínicos, Hospital Regional de Bata, Bata 88240, Equatorial Guinea; 5Grupo de Investigación en Microbiología Translacional, Instituto de Investigación Sanitaria del Principado de Asturias (ISPA), Microbiology Department, Hospital Universitario Central de Asturias (HUCA), 33011 Oviedo, Spain; 6Centro Nacional de Medicina Tropical (CNMT), Instituto de Salud Carlos III (ISCIII), 28029 Madrid, Spain; 7Centro de Investigación Biomédica en Red en Enfermedades Infecciosas (CIBERINFEC), 28029 Madrid, Spain; 8Centro de Investigación Biomédica en Red en Epidemiología y Salud Pública (CIBERESP), 28029 Madrid, Spain

**Keywords:** HIV-1, drug resistance, antiretrovirals, mutations, Equatorial Guinea, therapeutic failure

## Abstract

A lack of HIV viral load (VL) and HIV drug resistance (HIVDR) monitoring in sub-Saharan Africa has led to an uncontrolled circulation of HIV-strains with drug resistance mutations (DRM), compromising antiretroviral therapy (ART). This study updates HIVDR data and HIV-1 variants in Equatorial Guinea (EG), providing the first data on children/adolescents in the country. From 2019–2020, 269 dried blood samples (DBS) were collected in Bata Regional Hospital (EG) from 187 adults (73 ART-naïve/114 ART-treated) and 82 children/adolescents (25 HIV-exposed-ART-naïve/57 ART-treated). HIV-1 infection was confirmed in Madrid by molecular/serological confirmatory tests and ART-failure by VL quantification. HIV-1 *pol* region was identified as transmitted/acquired DRM, predicted antiretroviral susceptibility (Stanfordv9.0) and HIV-1 variants (phylogeny). HIV infection was confirmed in 88.1% of the individuals and virological failure (VL > 1000 HIV-1-RNA copies/mL) in 84.2/88.9/61.9% of 169 ART-treated children/adolescents/adults. Among the 167 subjects with available data, 24.6% suffered a diagnostic delay. All 125 treated had experienced nucleoside retrotranscriptase inhibitors (NRTI); 95.2% were non-NRTI (NNRTI); 22.4% had experienced integrase inhibitors (INSTI); and 16% had experienced protease inhibitors (PI). At sampling, they had received 1 (37.6%), 2 (32%), 3 (24.8%) or 4 (5.6%) different ART-regimens. Among the 43 treated children–adolescents/37 adults with sequence, 62.8/64.9% carried viruses with major-DRM. Most harbored DRM to NNRTI (68.4/66.7%), NRTI (55.3/43.3%) or NRTI+NNRTI (50/33.3%). One adult and one child carried major-DRM to PI and none carried major-DRM to INSTI. Most participants were susceptible to INI and PI. DRM was absent in 36.2% of treated patients with VL > 1000 cp/mL, suggesting adherence failure. TDR prevalence in 59 ART-naïve adults was high (20.3%). One-half (53.9%) of the 141 subjects with *pol* sequence carried CRF02_AG. The observed high rate of ART-failure and transmitted/acquired HIVDR could compromise the 95-95-95-UNAIDS targets in EG. Routine VL and resistance monitoring implementation are mandatory for early detection of ART-failure and optimal rescue therapy selection ART regimens based on PI, and INSTI can improve HIV control in EG.

## 1. Introduction

The human immunodeficiency virus (HIV) pandemic is still spreading in the human population worldwide. The Joint United Nations Programme on HIV and AIDS (UNAIDS) estimates that 38.4 million people were living with HIV (PLHIV) in 2021, with around 1.5 million new HIV infections and 650,000 AIDS-related deaths [1]. Equatorial Guinea (EG) is a small country located in West Central Africa between Cameroon and Gabon. HIV/AIDS is a serious public health problem in the country, being one of the most important causes of morbi-mortality in the population. The global HIV prevalence of 7.8% in EG is the highest within its Word Health Organization (WHO) region. Women represent almost 53% of adults (>15 years-old) living with HIV, with higher HIV prevalence than men (9.4% vs. 5.2%) [1,2]. The HIV prevalence in at-risk populations, such as sex workers, men who have sex with men, transgender people, injecting drug users or prisoners in the country, is still unknown. The most common HIV transmission routes are heterosexual, followed by vertical transmission from HIV-infected mother to child and blood transfusions [3,4]. Among the 66,000 PLHIV in the country, 8.6% are children and adolescents [2]. Adolescents are at high risk of acquiring the infection due to low knowledge about HIV infection, early start of sexual relations and the lack of prevention measures [3,4,5].

Improving access to HIV diagnosis and antiretroviral treatment (ART), along with implementing measures to prevent risk behaviors for disease transmission, is key to HIV control. ART has scaled up at an unprecedented rate over the past decade. At the end of 2020, 28.7 million PLHIV were receiving ART globally, reducing HIV-1 mortality and incidence [6]. By the end of 2003, the Government of the Republic of Equatorial Guinea started the regulation and implementation of ART through the National Plan against HIV/AIDS; diagnosis and treatment became 100% free of charge, guaranteeing access to the entire population. However, the coverage of people receiving ART in 2020 was 38% for adults (>15 years-old) and 28% for children (0–14 years-old), well below UNAIDS estimations for West and Central Africa (82% for adults and 35% for children) [2,3,4,5,6,7]. Furthermore, viral load (VL) and HIV drug resistance (HIVDR) monitoring for ART optimization are currently absent during the clinical routine in all patients, even those with therapeutic failure.

Despite the benefits of ART, HIVDR remains a major challenge to ART efficacy. Interruption of ARV exposure due to loss of patient follow-up, non-adherence to therapy or stock-outs affecting temporary drug availability can cause the emergence and the spreading of resistant variants. HIVDR can lead to regimen failure, limiting future therapy options [8,9]. Thus, the periodic monitoring of drug resistance mutations (DRM), selected when the virus replicates in the presence of sub-therapeutic levels of antiretroviral drugs (ARV), remains necessary. WHO recommends surveillance studies to detect the prevalence of transmitted (TDR) and acquired (ADR) DRM in ART-naïve (naïve) and ART-treated (treated) population, respectively [10,11]. Also, WHO recommends the pretreatment of HIVDR (PDR) prevalence monitoring. According to the WHO operational definition, PDR refers to HIVDR detected in ART-naïve subjects or previously ARV-exposed persons reinitiating first-line ART [12]. HIVDR data can guide clinicians towards first-ARV regimen selection in naïve patients or to optimization of second-line ART regimen in treated patients under virological failure. Genotypic resistance testing has not been implemented in routine care in most low and middle-income countries due to a lack of resources. EG, despite high HIV prevalence, ART scale-up and low adherence to ART [13], lacks surveillance studies to monitor HIVDR.

The use of ARV with a high genetic barrier may reduce the emergence of ADR, but their high prices limit their widespread use in low- and middle-income countries. In 2018, WHO published interim guidelines recommending the integrase inhibitors (INSTI) dolutegravir-containing regimens as the preferred first- and second-line ART regimens for PLHIV [14]. EG integrated this new drug into its first-line treatment in 2019 due to its advantages of effectiveness—easier to take, fewer side effects and higher genetic barrier for developing HIVDR than other drugs currently available in the country [15].

ART regimens based on non-nucleoside retrotranscriptase inhibitors (NNRTI) are the most frequently prescribed as first-line ART in many sub-Saharan African countries. If this continues, PDR to NNRTI is expected to exceed 10% in the region by 2030, leading to a significant extra burden of new AIDS deaths, new HIV infections and additional costs [16]. Individuals with PDR to NNRTI who initiate an NNRTI-based regimen are less likely to achieve VL suppression and more likely to have virological and adherence failure. This pattern is also observed in treated HIV-infected children [12]. The relatively high levels of PDR to nucleoside retrotranscriptase inhibitors (NRTI) among ARV-naïve infants suggest the need for caution when initiating regimens containing drugs with a low-genetic barrier to resistance. It also highlights the problems that the pediatric population will face, assuming the limited number of available drugs for pediatric use and the longer ART exposure compared to adults [17,18]. The data reinforce the need to fast-track the transition to dolutegravir-based first-line regimens and to use protease inhibitors (PI)-based ART when dolutegravir use is not feasible, following WHO recommendations [11].

Regarding HIV-1 variants, more than 140 have been described based on genetic homology: groups O, P, N and the epidemic group M. The latter causes more than 99% of nearly 38 million infections worldwide and is subdivided into 10 subtypes (A, B, C, D, F, G, H, J, K and L); more than 130 circulating recombinants forms (CRF, https://www.hiv.lanl.gov/content/sequence/HIV/CRFs/CRFs.html (accessed on 1 September 2022)); and innumerable unique recombinant forms (URF). Around 88% of PLHIV in resource-limited settings carry HIV-1 non-B subtypes and recombinants [19]. Genetic differences among HIV-1 variants can influence disease progression and the evolution of antiretroviral drug resistance [20]. Due to high genetic variability and viral evolution, HIV presents natural variant specific polymorphism, with some of them in positions related to drug resistance [21] that may affect the genetic barrier [22], resistance pathways [23], viral fitness [24] and the susceptibility to specific ARV [25]. HIV genetic variability can also lead to viral load underestimation or non-detection of viral RNA, affecting early molecular diagnosis and infection monitoring [26]. However, EG lacks regular surveillance studies for monitoring changes in the HIV-1 molecular epidemiology to track the circulating HIV-1 variants over time. In settings lacking resources for whole blood or plasma collection and handling, the use of dried blood samples (DBS) is a WHO-recommended alternative for VL monitoring and HIVDR and HIV-1 molecular epidemiology surveillance studies [27], being easier to collect, store and transport without the need for a cool chain [28].

The rate of HIVDR in EG is unknown so far, as resistance testing is not implemented in the routine clinic. A few studies have reported resistance data and information regarding HIV-1 subtypes and recombinants circulating in EG, but only in specific groups within the adult population using samples collected during 1997–2013 [29,30,31,32], with no data on pediatric and adolescent populations. These previous studies revealed a complex epidemic in the country, with CRF02_AG as the predominant variant like in the neighboring countries [19], and an alarming increase in resistance over time to the most widely used ARV. To update HIVDR data and the HIV-1 molecular epidemiology in the country, we analyzed the presence of DRM in the four main ARV classes (NRTI, NNRTI, PI and INSTI) and its therapeutic impact in the HIV-infected population under clinical care in Bata (EG) during 2019–2020, analyzing the circulating HIV-1 variants in the study population.

## 2. Materials and Methods

### 2.1. Ethics

This research was conducted by the Declaration of Helsinki and was approved by the local Ethics Committee for Clinical Investigation from Hospital Universitario Ramón y Cajal (Madrid, Spain) (ACTA 369, 24 June 2019). Signed informed consent was required for all adults and for all parents or legal guardians for children included in the study cohort.

### 2.2. Samples Collection

DBS were collected in EG during 2019–2020 from 269 subjects, including 187 HIV-infected adults (73 naïve, 114 treated), 57 HIV-infected children/adolescents (all treated) and 25 HIV-exposed children born to HIV-infected women, all under clinical follow-up in Bata Regional Hospital. All but one HIV-positive subject had been previously diagnosed in EG with rapid diagnostic testing Determine™ HIV-1/2 Ag/Ab (Abbott, Scarborough, ME, USA) and Bioline HIV-1/2 (Standard Diagnostics, Suwon, South Korea), following national guidelines [15]. Samples from all naïve patients in our cohort were collected during the first clinic visit after HIV diagnosis, where treatment was provided. According to clinical files, none of them took prior treatment. Therefore, all PDR should have been transmitted PDR. DBS were prepared by adding 70 µL of venous blood collected by venipuncture on a Whatman 903 Protein Saver Card (Cytiva, Little Chalfont, United Kingdom). They were dried overnight at room temperature, stored in a hermetically sealed bag with 2 desiccant bags and kept at −20 °C until transport on dry ice to the HIV-1 Molecular Epidemiology Laboratory in Ramón y Cajal University Hospital in Madrid, Spain, where the samples were stored at −80 °C until further analysis.

### 2.3. Confirmation of HIV Diagnosis and ART-Failure Identification

HIV-1 infection was confirmed in Madrid by molecular or serological confirmatory tests, depending on the patient’s age. The point-of-care Xpert^®^ HIV-1 Qual Assay (Cepheid, Sunnyvale, CA, USA) was used to confirm or discard the HIV infection in all 25 exposed infants (<18 months), which provides a binary “detected”/“not detected” result [33]. We confirmed HIV-1 infection in the remaining subjects under study when the VL was detectable (>40 HIV-1 RNA copies per plasma milliliter) after using the point-of-care Xpert^®^ HIV-1 Viral Load Assay (Cepheid, Sunnyvale, CA, USA) [34] in one DBS-dot eluted in Xpert Qualitative buffer, as previously reported [35]. We provided the number of HIV-1 RNA copies per dot and per plasma milliliter (cp/mL) after considering the patient’s hematocrit, assuming 33% hematocrit for children < 5 years-old; 39% for children > 5 years-old; 31.7% for pregnant women; 42% for non-pregnant women; and 47% for men, according to previous studies [36,37,38]. These hematocrits led to plasma volumes of 46.9 μL, 42.7 μL, 47.81 μL, 40.6 μL and 37.1 μL, respectively, in 70 μL blood collected per dot. ART-failure was considered in treated patients when VL was ≥ 1.000 cp/mL. In the samples with viraemia <40 cp/mL, we confirmed the infection with the Geenius HIV-1/2 Confirmatory Assay (BioRad, Hercules, CA, USA), using 1 DBS dot per patient eluted in elution buffer 3 (NucliSens EasyMag, BioMérieux, Marcy-l'Étoile, France), as previously published by our group [39].

### 2.4. Resistance Analysis

HIV-1 RNA was extracted from 2 DBS dots by automated magnetic silica extraction using the EasyMag extractor (BioMérieux, Marcy-l'Étoile, France) following elution with lysis buffer (BioMérieux, Marcy-l'Étoile, France). The extracted RNA was amplified by RT-PCR and nested PCR to obtain the HIV-1 *pol* coding region, using primers designed by WHO [27] for protease (PR) and reverse transcriptase (RT) amplification and ANRS primers for integrase (IN) amplification [40], as previously described [41,42]. PCR amplicons were purified with Illustra^TM^ ExoProStar^TM^ (Cytiva, Little Chalfont, United Kingdom) and sequenced by Macrogen Inc. Sequences were assembled and manually edited using Lasergene software. Viral sequences included the complete HIV-1 PR (codons 1–99), partial RT (1–345) and IN (50–288) for the genotyping study of DRM to NRTI, NNRTI and major and minor DRM to PI and INSTI. Major DRM to PI or INSTI is a nonpolymorphic DRM that makes a major contribution to reduced susceptibility to one or more PI or INSTI. Minor DRM to PI or INSTI are nonpolymorphic or minimally polymorphic mutations that contribute to reduced susceptibility in combination with major DRM “https://hivdb.stanford.edu/page/release-notes/#drug.resistance.mutations.drms.and.sequence.interpretation (accessed on 13 June 2022)”. DRM in the pretreated population were characterized by Stanford HIVdb Program v9.0 (Stanford University, Palo Alto, CA, USA) “https://hivdb.stanford.edu/hivdb/by-mutations/ (accessed on 13 June 2022)”, the algorithm also used to predict the resistance level to 25 ARV in *pol* genotypes. Since PDR could be TDR or ADR, we analyzed *pol* sequences from all ARV-naïve patients by the WHO TDR list 2009 [10] and by the Stanford algorithm v9.0. Both were available on the Stanford HIV website “https://hivdb.stanford.edu/ (accessed on 13 June 2022)”. The first tool is used for HIVDR surveillance for epidemiological purposes, and the second one is intended for clinical purposes. The WHO TDR list is implemented in the Calibrated Population Resistance (CPR) tool v8.0 “https://hivdb.stanford.edu/cpr/ (accessed on 13 June 2022)” and available in https://hivdb.stanford.edu/page/who-sdrm-list/ (accessed on 13 June 2022) for PR and RT and in https://hivdb.stanford.edu/page/insti-sdrm-list/ (accessed on 13 June 2022) for IN [43].

### 2.5. HIV-1 Variants’ Characterization

PR, RT and IN nucleotide sequences were aligned using the ClustalW algorithm implemented in MEGA6 to characterize HIV-1 variants. Phylogenetic trees were reconstructed as previously described [44], considering a branch support of >70%. Reference sequences of each HIV-1 group M subtype, sub-subtype and CRF were used to classify HIV-1 variants. Sequences not clustering with any known subtype or CRF were analyzed using the Recombination Detection Program (RDP3v4.13, Darren Martin), identifying the subtypes involved in eventual recombination events and hypothetical recombination breakpoints. To further confirm the detected putative recombination events, new phylogenetic analyses were performed using the sequence fragments assigned to different subtypes according to the proposed breakpoint position(s) defined by RPD3. In the positive cases, the recombinant sequences were redefined as URF.

### 2.6. Statistics Analysis

Medians were assessed for data not normally distributed. The statistical significance was calculated using the Fisher exact test or Chi-square for categorical variables and the Mann–Whitney test for continuous variables. Two-sided *p*-values of <0.05 were considered statistically significant. The percentage of viruses carrying DRM was calculated with 95% confidence intervals. Statistical analyses were conducted using GraphPad Prism v8.0.1 (GraphPad Software, San Diego, CA, USA).

### 2.7. Accession Numbers

HIV-1 sequences were submitted to GenBank with the following accession numbers: OL470728-OL470865 and OP512557-OP512559.

## 3. Results

### 3.1. Study Population

Among the 269 DBS collected in EG from 187 HIV-infected adults (73 naïve, 114 treated) and 82 children/adolescents (25 HIV-exposed, 57 HIV-infected and treated) under clinical follow-up in the Regional Hospital of Bata, we confirmed in Madrid the HIV-1 infection in 237 (88.1%) of them (Table 1). Among them, 40 were children (2 HIV-exposed, 38 treated); 18 adolescents (all treated); and 179 adults (66 naïve, 113 treated). Of the 25 HIV-exposed infants, we confirmed the absence of HIV-1 infection in 23 (92%), revealing an 8% [95% CI, 1.4–24.9] mother-to-child transmission rate. Among the 244 patients diagnosed as HIV-positive in EG using rapid diagnostic tests, we identified 9 (3.7%) subjects with false-positive HIV diagnosis in EG: 7 naïve adults, 1 treated adult and 1 treated child, 2 of them under unnecessary ART in EG. The median age at sampling of HIV-infected children, adolescents and adults was 6/14/34 years-old, respectively. Most of the 25 children (88%) and 8 adolescents (75%) with available data related to the HIV infection route were infected by mother-to-child transmission. The remaining 5 acquired the infection by transfusion, being 6, 11, 12 and 13 years-old at sampling, respectively. All 102 adults with a known route of HIV infection had acquired the virus by sexual transmission.

CD4 levels below 200 cells/mm^3^ at diagnosis in EG helped to estimate the diagnostic delay in the country, showing that 24.6% of the 167 subjects with known data suffered diagnostic delay (8% children/21.4% adolescents/28.1% adults). Among the 137 patients with recorded comorbidities in clinical files, 122 (89.1%) had from 1 to 4 comorbidities, and more than one-half (57.7%) presented only 1 (Table 1). Children presented significantly fewer comorbidities than adults.

### 3.2. ART-Failures and Treatment Delay in HIV-Infected and Treated Population

Among the 169 ART-experienced patients with confirmed HIV-1 infection, 69.8% were under therapeutic failure, showing viraemia greater than 1000 cp/mL at sampling. Therapeutic failure was significantly higher in children and adolescents than in adults (84.2% and 88.9% vs. 61.9%, *p* < 0.05) (Figure 1).

Among the 121 treated subjects with known ART initiation and first diagnosis dates, 19.8% suffered a delay in treatment start, with more than 1 year between first diagnosis and ART initiation: 13.5% in children, 17.6% in adolescents and 23.9% in adults (Table 1). Moreover, 2.7% of children, 11.8% of adolescents and 10.4% of adults under ART have experienced treatment delays of more than 3 years. Median time on ART at sampling of 121 treated children/adolescents/adults was 2.3/3.3/4.7 years, respectively.

All the 125 treated individuals with ART-experience information had received NRTI, 95.2%; NNRTI, 22.4%, were INSTI-experienced; and 16% were PI-experienced. The use of NNRTI was significantly higher in children than in adolescents (100% vs. 83.3%, *p* < 0.05), and the INSTI administration was higher in adults than in children/adolescents (35.7% vs. 5.4/5.6%, *p* < 0.05). At sampling, the 125 subjects were under their 1st (37.6%), 2nd (32%), 3rd (24.8%) or 4th (5.6%) different ART regimen. Seven patients (two children, two adolescents and three adults) had failed the treatment at least three times, being under the fourth regimen at sampling. Most participants had received lamivudine (94.4%), followed by efavirenz (80%), tenofovir (68%), zidovudine (57.6%) and nevirapine (50.4%). Adolescents and adults presented significantly higher exposure than children to tenofovir (83.3% and 90% vs. 18.9%, *p* < 0.0001) and emtricitabine (55.6% and 37.1% vs. 2.7%, *p* < 0.0001), while zidovudine was administered more frequently to children than adolescents and adults (100% vs. 61.1% and 34.3%, *p* < 0.001), along with nevirapine (70.3% vs. 44.4% and 41.4%, respectively, *p* < 0.05). The use of dolutegravir was significantly higher in adults than in children and adolescents (35.7% vs. 5.4% and 5.6%, *p* < 0.05) (Table 1).

### 3.3. DRM among 61 Naïve and 80 Treated Patients

HIV-1 *pol* sequence was recovered, in at least 1 region, from 141 (59.5%) patients (Appendix A) of which were 32 children (2 naïve, 30 treated), 13 treated adolescents and 96 adults (59 naïve, 37 treated), recovering 131PR/121RT/97IN sequences (Table 1 and Appendix A). All 80 (56.7%) treated subjects with available *pol* sequence were under therapeutic failure (VL > 1000 cp/mL).

DRM to the main four ARV-classes among naïve and treated populations with available *pol* sequence are shown in Figure 2 and Appendix A. Among the 141 naïve and treated subjects with sequence, 64 (45.4%) were infected with viruses carrying major-DRM to 1 (24.1%), 2 (20.6%) or 3 (0.7%) ARV classes. Of the 80 treated patients with available PR, RT or IN sequence, 51 (63.8%) carried resistant strains to 1 (26.3%), 2 (36.3%) or 3 (1.3%) ARV-classes. Considering this group, among the 43 children-adolescents/37 adults with *pol* sequence, 62.8/64.9% carried viruses with major-DRM, affecting 1 (16.3/37.8%) or 2 (44.2/27%) ARV-classes. Most of these 80 subjects harbored DRM to NNRTI (68.4/66.7%), NRTI (55.3/43.3%) or NRTI+NNRTI (50/33.3%), mainly affecting nevirapine/efavirenz/emtricitabine/lamivudine. Only 1 (2.3%) 13 year-old adolescent with 7 years of ART-experience presented triple resistance, being under a 2nd ART regimen at sampling (zidovudine+lamivudine+efavirenz and zidovudine+lamivudine+lopinavir/ritonavir). Major-DRM to the four drug classes (NRTI, NNRTI, PI, INSTI) were not found in any patient with available PR, RT and IN sequences. One adult and one child carried major-DRM to PI, and no subject carried viruses with major-DRM to INSTI. Minor-DRM to INSTI was found in 5 (9.3%) treated subjects. Two adults with major-DRM were pregnant women at sampling (one with DRM to NNRTI and one with NRTI+NNRTI).

Among the 61 naïve patients with sequence, 13 (21.3%, all adults) presented TDR only to 1 drug class: 9.6% to NNRTI, 3.9% to NRTI, and 2.4% to INSTI (APOBEC mutation). Thus, TDR prevalence in 59 naïve adults was high by Stanford (20.3%), revealing infection by resistant HIV in 1 out of 5 naïve adults. One naïve adult with resistant viruses was a pregnant woman at sampling. Neither of the two naïve children presented TDR.

We also identified DRM to the main ARV-classes presented in the 141 subjects with *pol* sequence, comparing treated vs. naïve patients (Figure 3 and Appendix A). Among the 68 treated patients with RT sequences, 34 (50%) carried DRM to NRTI, being the most frequent change was M184V/I (44.1%), followed by K219Q/N/E/R (16.2%), T215F/Y/P/SY (13.2%), K70R/E and M41L (10.3% each), while just 2 (3.8%) of 53 naïve patients with RT sequence presented DRM to this drug class. DRM to NNRTI were identified in 46 (67.6%) treated subjects, carrying K103N/H/S (47.1%) as the main DRM, in addition to G190S/A, A98G and V108I (16.2% each). On the other hand, 10 (18.9%) naïve patients carried resistant viruses to NNRTI, with V179E/D (11.3%) and K103N/H/S (7.5%) as the most prevalent DRM. Major-DRM to PI were presented in 2 treated patients (1 child and 1 adult) and minor-DRM to PI in 1 naïve adult patient DRM to INSTI was found in 5 (9.3%) treated subjects (4 adults and 1 child), all of them with minor-DRM, and in 4 (9.3%) naïve adult individuals with major-DRM (2.3%) and minor-DRM (7%). The only major-DRM to INSTI present in a naïve adult was E92K, an APOBEC mutation identified as major-DRM to INSTI by Stanford “https://hivdb.stanford.edu/page/apobecs/ (accessed on 15 July 2022)”. Albeit this mutation is associated with a resistance site, it is an unusual mutation that does not confer resistance.

All 65 treated participants (42 children/adolescents and 23 adults) with *pol* sequence and available ART regimen data were NRTI-experienced, 63 (96.9%) had received NNRTI, 13 (18.5%) PI and 10 (15.4%) INSTI. However, all the 80 treated patients (43 children/adolescents and 37 adults) with *pol* sequence were under therapeutic failure (>1000 cp/mL) at sampling. According to the found resistance data, in 63.8% (62.8% children/adolescents and 64.9% adults) of them, the lack of viral suppression was due to resistant viruses. DRM was absent in 36.2% (37.2% children/adolescents and 35.1% adults) of treated patients with unsuppressed viraemia (>1000 cp/mL), suggesting adherence failure (Appendix A).

### 3.4. Transmitted Pretreatment Drug Resistance Mutations by Different Tools

Transmitted pretreatment drug or PDR prevalence in the 59 naïve adults with *pol* sequence was analyzed by both Stanfordv9.0 and the CPRv8.0 tool (WHO TDR list 2009) (Figure 4 and Figure 5). The two naïve children were excluded from the analysis as they did not carry viruses with major or minor-DRM.

#### 3.4.1. Transmitted PDR Mutations Analysis with the Cprv8.0 Tool (WHO TDR List 2009)

Among the 59 naïve adults, the total TDR prevalence by WHO TDR list 2009 was 10.2%. The WHO TDR list 2009 did not identify some of the DRM to NRTI and NNRTI and minor-DRM to PI and INSTI, since that list did not include all DRM to RT inhibitors and no minor-DRM, such as E44D (NRTI) and V106I, E138A and V179 (NNRTI) [10].

#### 3.4.2. Transmitted PDR Mutations Analysis with the Stanfordv9.0 Tool

The total PDR prevalence by this tool was 20.3% in the 59 naïve adults, observing DRM such as M41L (NRTI), K103N (NNRTI) and G190A (NNRTI). Minor-DRM to PI (L89V) and INSTI (E157Q, G163K) were only identified by this tool and not by the WHO TDR list 2009.

### 3.5. Mother-to-Child DRM Transmission in the Study Cohort

Among the 80 treated patients with *pol* sequence, 6 were HIV-infected mother-child pairs under ART with unsuppressed viraemia. The six mother-child pairs had PR and RT sequences, and only two of them were IN sequences. Table 2 shows the HIV-1 variants and DRM found in each mother-child pair. All mothers and children carried non-B variants (sub-subtype A3, subtype G, CRF02_AG, CRF37_cpx, URF_02BA), and 4 (66.7%) mother-child pairs carried the same HIV-1 variant. All six mothers and six children presented major-DRM to NRTI, NNRTI or PI. In three mother-child pairs (P1, P2 and P6), the child presented at least one DRM observed in the corresponding mother to NRTI and/or NNRTI (M184V+K103, M41L and K103N, respectively). One mother (P4 pair) carried major-DRM to PI (M46L) non-transmitted to her child. Just one minor-DRM to INSTI (T97A) was found in one of the mothers (P6 pair), but since the IN sequence of the child was unrecovered, we could not see if they shared that mutation.

### 3.6. Predicted Antiretroviral Susceptibility

Prediction of antiretroviral susceptibility to 25 ARV by Stanfordv9.0 was performed in the 141 HIV-1 infected patients of EG with available *pol* sequence (Figure 6 for the total population with *pol* sequence; Appendix A for individual patients). Most treated and naïve subjects were infected with viruses susceptible to PI (97.7%) and INSTI (91.8%). Considering naïve/treated subjects, susceptibility to NRTI was observed in 98.1/50%, susceptibility to NNRTI in 79.2/32.4%, susceptibility to PI in 98.3/97.2% and susceptibility to INSTI in 93/90.7%.

When high and intermediate resistance levels were considered, 42.1% of subjects with *pol* sequence carried resistant viruses to NNRTI, 26.4% to NRTI and 0.8% to PI. None of the identified viruses presented high/intermediate resistance to INSTI. The most affected NNRTI were nevirapine (42.1%), efavirenz (39.7%) and doravirine (20.7%). When considering NRTI, 25.6% of the sequences presented high/intermediate resistance to emtricitabine and lamivudine, while 9.1% were resistant to abacavir, stavudine and didanosine. Significant differences were found between treated and naïve groups with high/intermediate resistance to NNRTI (67.6% vs. 9.4%; *p* < 0.0001) and NRTI (47.1% vs. 0%; *p* < 0.0001) and in susceptibility to NNRTI (32.4% vs. 79.2%) and NRTI (50% vs. 98.1%).

### 3.7. HIV-1 Viral Variants

To update the HIV-1 molecular epidemiology in EG, HIV-1 variants were characterized by phylogeny in the available *pol* sequences (Figure 7). We observed that most (85.9%) of the circulating variants in the 141 subjects with *pol* sequence collected during 2019–2020 were recombinants (URF, CRF), mainly CRF (66%). Indeed, 2 out of 3 subjects in the study cohort were infected with 7 different CRFs, being CRF02_AG the most prevalent CRF (81.7%) and the HIV-1 variant (53.9%). Only 20 (14.1%) patients carried HIV-1 pure subtypes at *pol* (6A, 1B, 5C, 1D, 1F, 4G, 1H, 1K), the remaining viruses (20%) complex URF.

## 4. Discussion

There is a lack of updated data on HIV resistance and circulating HIV-1 variants molecular epidemiology in EG. This study shows the most extensive and recent data regarding the DRM in the HIV-infected and -treated adult general population, providing the first DRM data for INSTI in infected children and adolescents and TDR in naïve subjects in the country. Previous similar studies in EG were conducted only on treated adult population with samples collected from 1999–2013, while our study included samples collected from 2019–2021 (Appendix A). We also provide the most recent data on the HIV-1 variants currently circulating in EG to update the HIV-1 epidemiology in the country.

Since 2016, WHO recommends immediate ART initiation for all PLHIV, regardless of CD4 count or clinical stage, as part of the “Treat-All” strategy [45]. Despite free access to HIV diagnosis and treatment in EG since 2004, AIDS-related deaths have increased by 44% in the last 12 years [46]. Furthermore, the country is far from reaching the 95-95-95% UNAIDS targets by 2030 since only 51% of PLHIV know their serological status, 81% of these are on ART and it is not known how many PLHIV under ART are virally suppressed [47]. Our results have revealed that more than one-half (69.8%) of treated subjects under study were not controlling the infection since they had VL > 1000 cp/mL at sampling.

Early detection of HIV infection is also critical to control the HIV spread, and an early ART and CD4 cell count can be used to estimate the diagnostic delay, assuming that CD4 levels in naïve HIV-infected patients decrease over time [48,49]. In our study, we found that 24.6% of the study population had a diagnostic delay, in line with neighboring countries [50]. The observed lower diagnostic delay in children vs. adolescents and adults (8% vs. 21% and 28%, respectively) would suggest that the HIV pediatric infection is being diagnosed at an increasingly earlier stage. To reduce the diagnostic delay in the country, efforts should be directed to HIV surveillance among children, adolescents and adult populations, facilitating access to sexual and reproductive health services for women and girls or implementing comprehensive programs in school and in out-of-school settings, involving the whole community. In EG, HIV monitoring is limited due to a weak public health system. The absence of HIV molecular diagnosis in the clinical routine in EG does not allow the early diagnosis of HIV-exposed children born to HIV-infected women, which is essential to prevent the rapid progression of the disease in a country with a high (25%) mother-to-child HIV transmission rate [51]. The only diagnostic option available for children < 18 months is repeated rapid serological testing, which increases the risk of false HIV diagnoses and infant mortality if not treated in time [52]. The 2 (8%, 2/25) HIV-exposed children who were infected died before they could receive a confirmatory diagnosis and adequate ART. The perinatal transmission rate found in our cohort (8%) was high but below the 21% reported by UNAIDS for the West and Central African Region [53], and slightly higher than other countries with well-implemented national elimination of vertical transmission programs [54,55]. However, the rate of HIV-1 transmission in EG has increased 3 times compared to recent years (8% [95% CI, 1.4–24.9] in 2019–2020 vs. 2.6% in 2012–2013) [56], assuming lower participation of Equatoguinean pregnant women in the prevention of mother-to-child transmission of HIV (PMTCT) programs. It reveals the urgency to reinforce the PMTCT programs in the country and the need for periodic surveillance studies that include a higher number of pregnant women, as well as the need to increase the number of HIV-exposed infants in further similar studies. Unfortunately, the high number of losses in clinical follow-up in HIV-exposed children in EG with confirmed negative diagnosis in Spain did not allow us to know how many of them seroreverted after sampling during the first 18 months of life.

At a global level, while 81% of pregnant women living with HIV and 76% of adults overall were receiving ARV in 2021, only 52% of children (0–14 years-old) and 55% of adolescents (15–19 years-old) were accessing ART, making us far from ending new HIV infections in these groups [57]. For ART monitoring and early detection of treatment failure, the WHO recommends VL testing. In countries without access to VL, such as in EG, DBS from a capillary or venous whole blood can be used for HIV monitoring, including early infant diagnosis, HIV resistance testing and viraemia quantification to identify ART-failures when VL is higher than 1000 cp/mL [27,58]. The causes of the up to three years delay in starting ART initiation after the HIV-positive diagnosis observed in some subjects under study were unknown, but they could be the patient’s perception of being healthy, the difficult access to treatment and the fear of stigma and discrimination by the community.

We observed that therapeutic failure was significantly higher in children and adolescents than in adults (84.2% and 88.9% vs. 61.9%, *p* < 0.05), accordingly to previous studies [59], due to adherence failures, suboptimal blood drug levels, inadequate regimens or infections with resistant viruses. Around one-third of the HIV-infected children in the world present virological failure within two years of ART [60]. The use of ART in HIV-infected mothers and prophylaxis in HIV-exposed newborns to prevent vertical transmission of HIV can also benefit the selection of viruses carrying DRM to ARV in infected neonates [61,62,63]. In our study, DRM was vertically transmitted in three of six mother/child pairs, revealing again the high importance of adequate ART and VL control in pregnant women. All mothers of the six pairs were on treatment at delivery time, but none had received ART during pregnancy; therefore, we assume that they were not enrolled in any PMTCT program. Only one out of six infants had received treatment (zidovudine) as prophylaxis at delivery. Since the 6 children had been on ART for an average of 2.4 years at sampling, we cannot distinguish if DRM were transmitted through mothers or acquired during the years on treatment. VL and resistance periodic monitoring should be encouraged in all subjects with DRM in the study, mainly in infants since the development of resistance at birth will condition the availability of therapy throughout their lives, as well as its effectiveness.

The identification of DRM is crucial to publish first and rescue-line ART guidelines at the country level and offering appropriate treatment to each patient. However, resistance monitoring is not routinely available in EG, and ARV regimen changes are determined by ARV availability, sometimes restricted by stock-outs, toxicity and secondary effects in patients. Nevertheless, DRM identification should drive any ART change in patients, especially in the younger infected population, as they are patients with a lifelong treatment and for more years than adults. Most patients in our study had experienced at least 1 ART regimen change before sampling, and even 5.4/11.1/4.3% of children/adolescents/adults had received 4 different ART regimens at sampling without any prior resistance study.

Major DRM prevalence in our ART-experienced patients showed that two-thirds (63.8%) of subjects with available *pol* sequence were infected with resistant strains, affecting mainly nevirapine and efavirenz (NNRTI) and emtricitabine and lamivudine (NRTI) susceptibility. Since most naïve and treated subjects under study in EG were infected with viruses susceptible to PI (97.7%) and INSTI (91.8%), dolutegravir and PI use in first and rescue regimens could improve VL suppression in naïve and treated patients, thereby reducing mortality and HIV incidence compared to NNRTI-based regimens.

According to both WHO TDR list 2009 naïve patients presented a 10.2% prevalence. This is of concern and could continue to increase if the transition to dolutegravir-based first-line regimens is not accelerated, as recommended by WHO when the prevalence of PDR among adults initiating first-line ART is ≥ 10% [12]. When comparing with previous HIVDR studies in EG [29,30,32] (Appendix A), we observed no significant increase in TDR in the last 12 years: 4.8% in 2008 [30] vs. 9.8% in the present study (2019–2020), especially to NNRTI (from 0 vs. 9.4%, *p* < 0.05) due to their growing use in ART regimens in the country over the years. We also detected a worrying and significant increase of DRM in treated individuals, ranging from 29% in Equatoguinean immigrants living in Spain during 1997–2011 [32] and 20% in pregnant women in EG during 2012–2013 to 63.8% in the general population in EG from 2019–2020 (this study), *p* < 0.0001), being observed a significant increase for NRTI, NNRTI and PI (Appendix A). These results reinforce the need for further HIVDR studies in the general population in the country to monitor HIVDR tendency in the future.

Regarding specific drugs, HIV treatment guidelines recommend oral INSTIs as the preferred ARV for individuals initiating therapy due to their efficacy, safety and ease of use [64]. Recently a clinical trial showed that dolutegravir-based therapy was superior to the standard of care in children and adolescents as first- and second-line therapy [65]. Infants < 25 kg guidelines recommend zidovudine + lamivudine + nevirapine or efavirenz in EG [15]. Per the WHO recommendation in 2018 [14], the last guidelines from ART in EG recommended the TLD combo (tenofovir + lamivudine + dolutegravir) as the first-line regimen and TLE combo (tenofovir + lamivudine + efavirenz) for second-line ART in children with > 25 kg, adolescents and adults [15]. However, at sampling, only 22.4% of treated subjects with ART information in our study cohort had experience with dolutegravir, whereas 65.6% were receiving NNRTI-based regimens, including nevirapine or efavirenz. We found that dolutegravir use was significantly higher in adults than in children and adolescents (35.7% vs. 5.4% and 5.6%, *p* < 0.05).

Our results provide the most recent data regarding HIV-1 molecular epidemiology in EG (Appendix A), where CRF02_AG is still the most prevalent variant, as it has been for the last 2 decades [29,32], causing about 54% of infections in our study cohort. However, a significant increase in infections caused by recombinants has been observed, rising from 56.1% during 1997–2011 [32] to 76.3% during 2012–2013 [29] and 85.8% (present study, 2019-2020), decreasing the prevalence of HIV-1 pure subtypes, also increasing 7 times the prevalence of complex and unique recombinants (URF), comparing the first and last study period (Appendix A). We also described the first subtype K in EG. These results reinforce the importance of periodic HIV-1 molecular epidemiology studies in the country, as HIV genetic variability could affect disease progression [66], ART efficacy of certain ARV [25,67], and the performance of molecular techniques, underestimating the VL or even failing to detect viral RNA [26].

This study presents some limitations. The first one was that we could not estimate the TDR prevalence in children due to the small number of naïve HIV-infected infants under study (only two). Thus, further analyses are required to monitor the current transmission of drug-resistant strains in naïve HIV-infected children and adolescents in EG. The second limitation was the absence of ART data from mothers of the HIV-infected children and adolescents in clinical files provided by doctors at the Bata Regional Hospital, which could have helped to identify cases of vertical DRM transmission and to compare their ART regimens and DRM prevalence with non-pregnant women and male adults. The last limitation would be the absence of resistance data due to negative PCR amplification caused by low viraemia in all patients with <1000 cp/mL and mainly to genetic variability in the viral target of the primers used for PCR in one-third of patients with detectable viraemia (>1000 cp/mL), as unique and complex recombinant forms have been increasing over time in EG [29,32], reaching almost 86% in our study.

## 5. Conclusions

This work provides the most comprehensive and recent information to date on HIV-1 resistance mutations, antiretroviral susceptibility and circulating variants in EG, where 7.8% of the population live with HIV [2]. In addition, the first data on the molecular epidemiology of HIV-1 in the pediatric population has been obtained. Our data reinforce the need to implement VL quantification and resistance monitoring during routine clinical practice, to strengthen the country’s laboratory services and to use INSTI- and PI-based regimes as an alternative to NNRTIs due to the high presence of DRM in this ARV-class. Furthermore, the study results encourage the need for a faster ART intervention after HIV diagnosis in EG to reduce the observed ART delay in the study cohort, where 1 in 10 HIV-infected adolescents and adults experienced treatment delays of up to more than 3 years, being much lower (2.7%) in children. All these actions would reduce the spread of resistant viruses in the population and help control HIV infection in EG to achieve the UNAIDS’s 95-95-95 target [68]. Continuous HIV-1 molecular epidemiological surveillance would help to identify the introduction of new variants in the country.

## Figures and Tables

**Figure 1 viruses-15-00027-f001:**
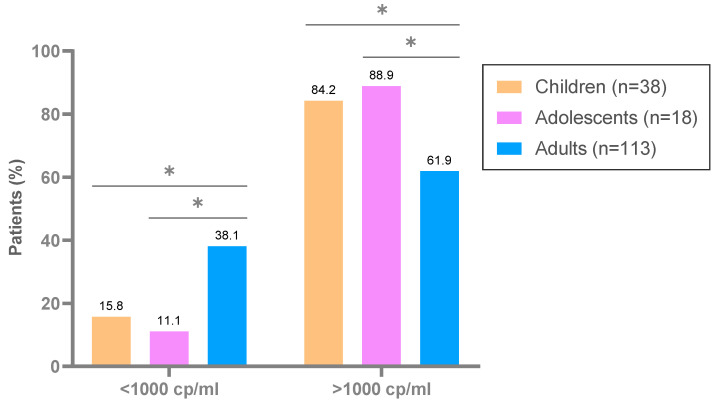
Therapeutic failure in the 169 HIV-infected and treated subjects of EG (2019–2020). Treated, ART-treated; cp/mL, copies of HIV-1 RNA per plasma milliliter; therapeutic failure, >1000 cp/mL; *, significant *p* values <0.05.

**Figure 2 viruses-15-00027-f002:**
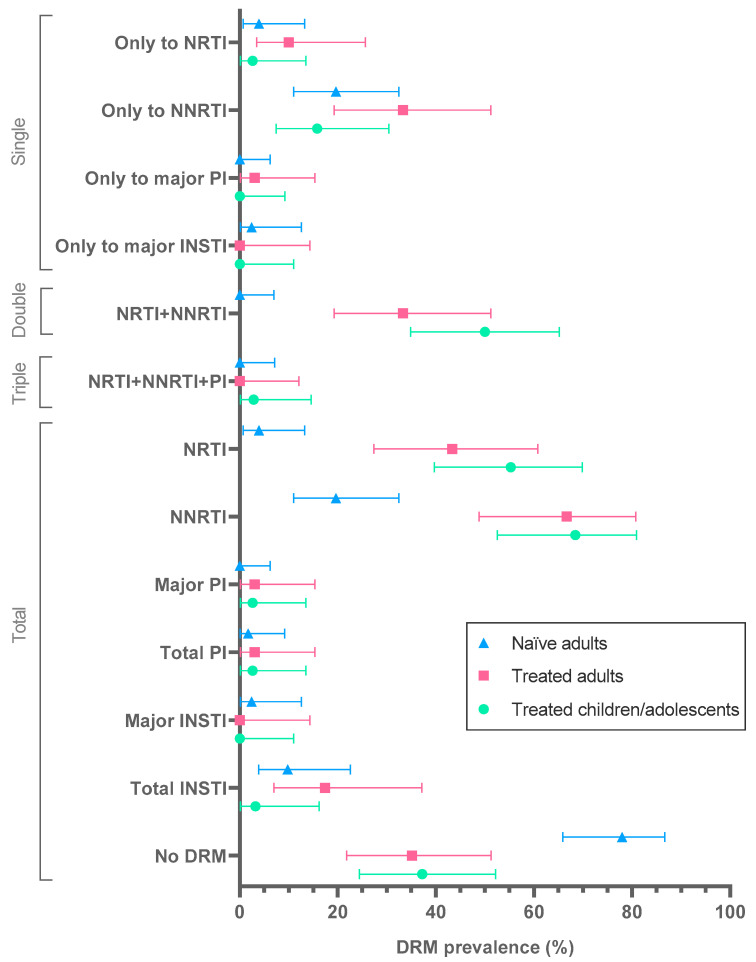
Percentage of HIV-infected patients carrying DRM to the main antiretroviral classes in Equatorial Guinea (2019–2020). Mean DRM prevalence (colored figures) and 95% confidence intervals in 139 subjects under study with available *pol* sequence after excluding the 2 naïve children since they did not present DRM: 129PR/119RT/95IN. Naïve, ART-naïve; treated, ART-treated; single resistance, to 1 ARV-class; double resistance, to 2 ARV-classes; triple resistance, to 3 ARV- classes; total resistance, by ARV-class; no DRM, no major-DRM found in the available sequenced *pol* regions per patient. DRM to PI and INSTI are always major unless indicated otherwise. DRM, drug resistance mutations; ARV, antiretrovirals; NRTI, nucleoside reverse transcriptase inhibitors; NNRTI, non-NRTI; PI, protease inhibitors; INSTI, integrase strand transfer inhibitors. Percentages were calculated over the number of sequences available from each region. More data available inAppendix A.

**Figure 3 viruses-15-00027-f003:**
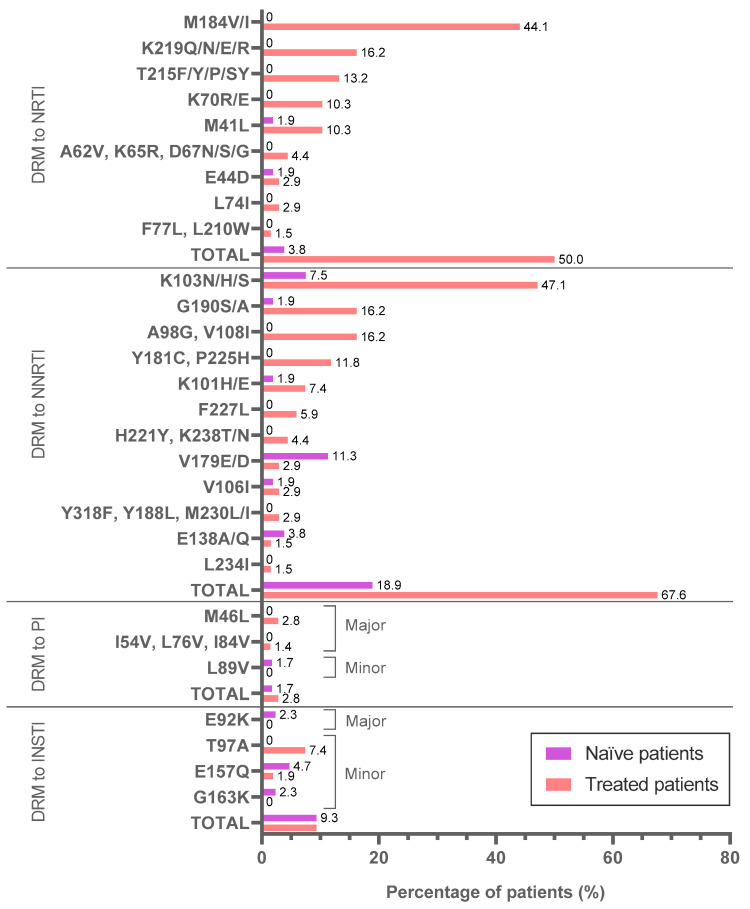
Drug resistance mutations to the main antiretroviral classes in the study cohort. Available sequences in 141 subjects under study: 131PR, 121RT and 97IN. DRM, drug resistance mutation; naïve, ART-naïve; treated, ART-treated; NRTI, nucleoside reverse transcriptase inhibitors; NNRTI, non-NRTI; PI, protease inhibitors; INSTI, integrase strand transfer inhibitors. We considered DRM to NRTI and NNRTI and major and minor-DRM to PI and INSTI, according to Stanford v9.0. DRM in the same line are equally prevalent. DRM to INSTI E92K is an APOBEC mutation identified as a major-DRM by Stanford. Percentages were calculated over the number of sequences available from each region. More data available in Appendix A.

**Figure 4 viruses-15-00027-f004:**
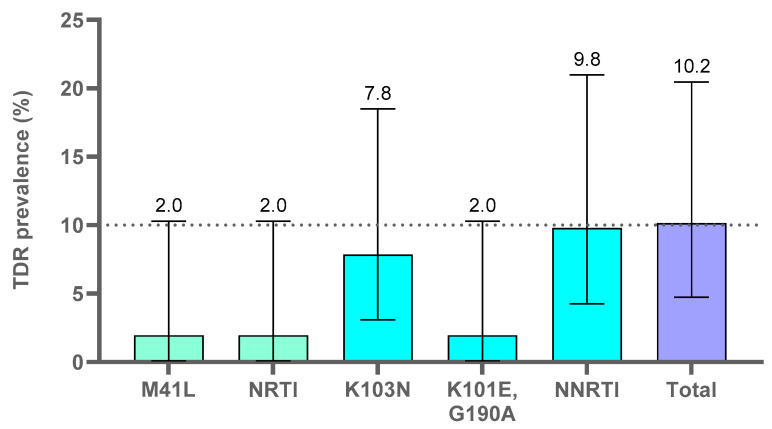
Transmitted pretreatment drug resistance mutations identified by WHO TDR list 2009 in the 59 naïve adults. Available sequences in 59 naïve adults of EG: 58PR, 51RT, 41IN (see Appendix A). TDR, transmitted drug resistance mutations; naïve, ART-naïve. NRTI, nucleoside reverse transcriptase inhibitors; NNRTI, non-NRTI; total, total TDR found in the available *pol* coding regions. We considered only major-DRM to PI, INSTI, NRTI and NNRTI, according to WHO TDR list 2009 “https://hivdb.stanford.edu/page/who-sdrm-list/ (accessed on 15 July 2022)” for PR and RT and “https://hivdb.stanford.edu/page/insti-sdrm-list/ (accessed on 15 July 2022)” for IN.

**Figure 5 viruses-15-00027-f005:**
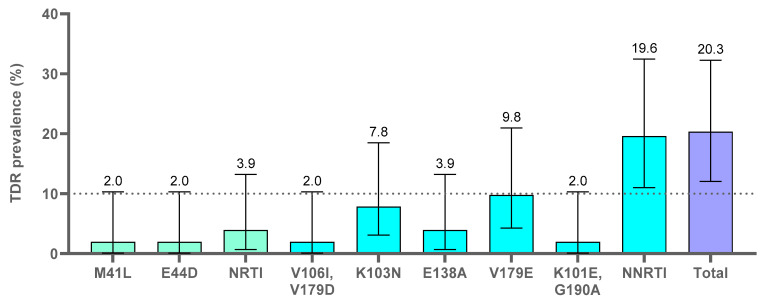
Transmitted pretreatment drug resistance mutations identified by Stanfordv9.0 in the 59 naive adults. Available sequences in 59 naïve adults of EG: 58PR, 51RT, 41IN (see Appendix A). Percentages were calculated, excluding the two naïve children since they did not carry viruses with major or minor-DRM. TDR, transmitted drug resistance mutations; naïve, ART-naïve. NRTI, nucleoside reverse transcriptase inhibitors; NNRTI, non-NRTI; total, total TDR found in the available *pol* coding regions. We considered only major-DRM to PI, INSTI, NRTI and NNRTI, according to Stanford v9.0. DRM to IN E92K was not included in this figure as it appears in the APOBEC mutation list provided by Stanford “https://hivdb.stanford.edu/page/apobecs/ (accessed on 15 July 2022)” and it does not confer resistance to any antiretroviral.

**Figure 6 viruses-15-00027-f006:**
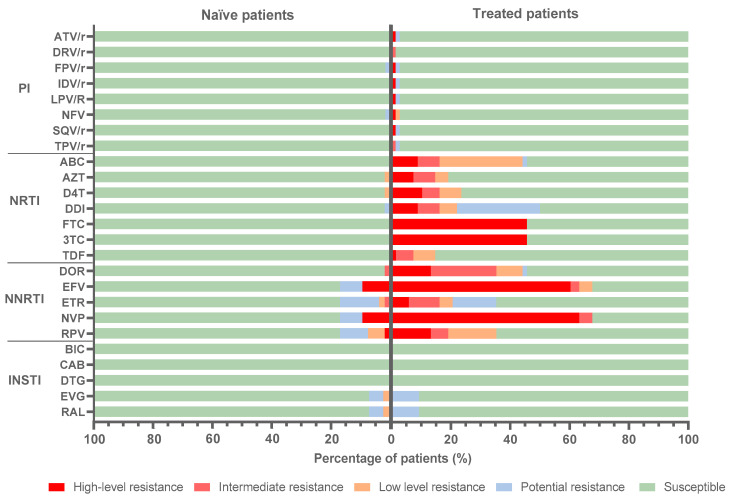
Predicted antiretroviral susceptibility in HIV-1 infected subjects in EG with available *pol* sequence from samples collected during 2019-2020. Data according to Stanfordv9.0 from 141 subjects with available sequences (131PR/121RT/97IN), 61 naïve (60PR/53RT/43IN) and 80 treated (71PR/68/RT/54IN), according to Appendix A. Naïve, ART-naïve; treated, ART-treated; NRTI, nucleoside reverse transcriptase inhibitors; NNRTI, non-NRTI; PI, protease inhibitors; INSTI, integrase strand transfer inhibitors. ATV/r, atazanavir/ritonavir; DRV/r, darunavir/ritonavir; FPV/r, fosamprenavir/ritonavir; IDV/r, indinavir/ritonavir; LPV/r, lopinavir/ritonavir; NFV, nelfinavir; SQV/r, saquinavir/ritonavir; TPV/r, tipranavir/ritonavir; ABC, abacavir; AZT, zidovudine; D4T, stavudine; DDI, didanosine; FTC, emtricitabine; 3TC, lamivudine; TDF, tenofovir; DOR, doravirine; EFV, efavirenz; ETR, etravirine; NVP, nevirapine; RPV, rilpivirine; BIC, bictegravir; CAB, cabotegravir; DTG, dolutegravir; EVG, elvitegravir; RAL, raltegravir. Found intermediate and high resistance: 42.1% to NNRTI (48/121), 26.4% to NRTI (32/121) and 0.8% to PI (1/131). Information per patient in Appendix A.

**Figure 7 viruses-15-00027-f007:**
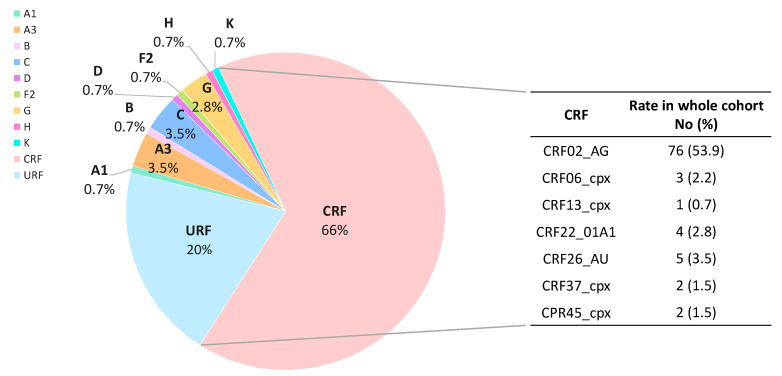
HIV-1 variants in Equatorial Guinea during 2019–2020. HIV-1 variants characterized in 141 infected subjects with available *pol* sequences (131PR/121RT/97IN) from samples collected during 2019–2020 in Equatorial Guinea. No, number; CRF, circulating recombinant forms; URF, unique recombinant forms.

**Table 1 viruses-15-00027-t001:** Epidemiological and virological characteristics of the 237 individuals from EG with confirmed HIV-1 diagnosis in Madrid.

	1	2	3	4	*P* Value		
Epidemiological and Clinical Features	Children (<12 y) No. (%)	Adolescents (12 y–17 y) No. (%)	Adults (≥ 18 y) No. (%)	Total cohort No. (%)	1 vs. 2	1 vs. 3	2 vs. 3
**Whole Study Cohort (*n* = 237)**							
**Total**	40 (16.9)	18 (7.6)	179 (75.5)	237 (100)			
**Female**	26 (65)	9 (50)	140 (78.2)	175 (73.8)	ns	ns	*
**Median age, years [range]**	6 (0.6–11)	14 (12–17)	34 (17–64)	30 (0.6–64)	***	***	***
**Route of transmission**							
**Vertical**	22 (55)	6 (33.3)	0	28 (11.9)	ns	***	***
**Sexual**	0	0	102 (57)	102 (43)	ns	***	***
**Transfusion**	3 (7.5)	2 (11.1)	0	5 (2.1)	ns	*	*
**Unknown**	15 (37.5)	10 (55.6)	77 (43)	102 (43)			
**ARV exposure**							
**Exposed/naïve**	2 (5)	0	66 (36.9)	68 (28.7)	ns	***	**
**Under ART**	38 (95)	18 (100)	113 (63.1)	169 (71.3)	ns	***	**
**Delayed diagnosis**							
**<200 cells/mm^3^**	2 (5)	3 (16.7)	36 (20.1)	41 (17.3)	ns	*	ns
**>200 cells/mm^3^**	23 (57.5)	11 (61.1)	92 (51.4)	126 (53.2)	ns	*	ns
**Unknown**	15 (37.5)	4 (22.2)	51 (28.5)	70 (29.5)			
**Comorbidities**							
**0**	11 (27.5)	1 (5.5)	3 (1.7)	15 (6.3)	ns	***	ns
**1**	2 (5)	3 (16.7)	74 (41.3)	79 (33.3)	ns	*	ns
**2**	0	0	33 (18.4)	33 (13.9)	ns	*	ns
**3**	0	0	9 (5)	9 (3.8)	ns	ns	ns
**4**	0	0	1 (0.6)	1 (0.5)	ns	ns	ns
**Unknown**	27 (67.5)	14 (77.8)	59 (33)	100 (42.2)			
**Subjects with available *pol* HIV-1** **Sequence (*n* = 141)**	**32 (80)**	**13 (72.2)**	**96 (53.6)**	**141 (59.5)**			
**PR**	29 (90.6)	11 (84.6)	91 (94.8)	131 (92.9)			
**RT**	30 (93.8)	10 (76.9)	81 (84.4)	121 (85.8)			
**IN**	24 (75)	9 (69.2)	64 (66.7)	97 (68.8)			
**HIV-1 variants (*n* = 141)**							
**Subtype B**	1 (3.1)	0	0	1 (0.7)	ns	ns	ns
**Non-B variants**	31 (96.9)	13 (100)	96 (100)	140 (99.3)	ns	ns	ns
**Art-Treated Patients at Sampling (*n* = 169)**							
**HIV-1 viral load**							
**<1000 cp/mL**	6 (15.8)	2 (11.1)	43 (38.1)	51 (30.2)	ns	*	*
**>1000 cp/mL**	32 (84.2)	16 (88.9)	70 (61.9)	118 (69.8)	ns	*	*
**Delayed ART**	**5 (13.5)**	**3 (17.6)**	**16 (23.9)**	**24 (19.8)**			
**Immediate^#^**	32 (84.2)	12 (66.6)	37 (32.7)	81 (47.9)	ns	*	ns
**<1 year**	0	2 (11.1)	14 (12.4)	16 (9.5)	ns	*	ns
**1–3 years**	4 (10.6)	1 (5.6)	9 (8)	14 (8.3)	ns	ns	ns
**>3 years**	1 (2.6)	2 (11.1)	7 (6.2)	10 (5.9)	ns	ns	ns
**Unknown**	1 (2.6)	1 (5.6)	46 (40.7)	48 (28.4)			
**Number of ART regimens at sampling**							
**1**	15 (39.5)	6 (33.3)	26 (23)	47 (27.8)	ns	ns	ns
**2**	12 (31.6)	5 (27.8)	23 (20.4)	40 (23.7)	ns	ns	ns
**3**	8 (21)	5 (27.8)	18 (15.9)	31 (18.4)	ns	ns	ns
**4**	2 (5.3)	2 (11.1)	3 (2.7)	7 (4.1)	ns	ns	ns
**Unknown**	1 (2.6)	0	43 (38)	44 (26)			
**Median time under ART in 121 ^‡^ treated subjects, years [range]**	2.3 (0.1–9.9)	3.3 (0.3–10.4)	4.7 (0.1–15.5)	3.6 (0.1–15.5)	ns	ns	ns
**NRTI experience ^†^**	**37 (100)**	**18 (100)**	**70 (100)**	**125 (100)**			
**AZT**	37 (100)	11 (61.1)	24 (34.3)	72 (57.6)	**	***	ns
**3TC**	36 (97.3)	17 (94.4)	65 (92.9)	118 (94.4)	ns	ns	ns
**TDF**	7 (18.9)	15 (83.3)	63 (90)	85 (68)	***	***	ns
**FTC**	1 (2.7)	10 (55.6)	26 (37.1)	37 (29.6)	***	***	ns
**D4T**	5 (13.5)	3 (16.7)	19 (27.1)	27 (21.6)	ns	ns	ns
**ABC**	3 (8.1)	0	0	3 (2.4)	ns	*	ns
**NNRTI experience ^†^**	**37 (100)**	**15 (83.3)**	**67 (95.7)**	**119 (95.2)**	*	ns	ns
**EFV**	29 (78.4)	13 (72.2)	58 (82.9)	100 (80)	ns	ns	ns
**NVP**	26 (70.3)	8 (44.4)	29 (41.4)	63 (50.4)	ns	*	ns
**PI experience ^†^**	**3 (8.1)**	**4 (22.2)**	**13 (18.6)**	**20 (16)**	ns	ns	ns
**LPV/r**	3 (8.1)	4 (22.2)	13 (18.6)	20 (16)	ns	ns	ns
**INSTI experience ^†^**	**2 (5.4)**	**1 (5.6)**	**25 (35.7)**	**28 (22.4)**	ns	*	*
**DTG**	2 (5.4)	1 (5.6)	25 (35.7)	28 (22.4)	ns	*	*

No, number; y, years; ARV, antiretrovirals; ART, antiretroviral treatment; VL, viral load; cp/mL, copies of HIV-1 RNA per plasma milliliter; cells/mm^3^, CD4 cells per cubic millimeter of blood; immediate^#^, less than a month; NRTI, nucleoside reverse transcriptase inhibitors; NNRTI, non-NRTI; PI, protease inhibitors; INSTI, integrase strand transfer inhibitors; AZT, zidovudine; 3TC, lamivudine; TDF, tenofovir; FTC, emtricitabine; D4T, stavudine; ABC, abacavir; EFV, efavirenz; NVP, nevirapine; LPV/r, lopinavir/ritonavir; DTG, dolutegravir; PR, protease; RT, reverse transcriptase; IN, integrase. ^†^, percentage of ARV experience was calculated according to known ART data in 125 subjects: 37 children, 18 adolescents, 70 adults. ns, not significant (*p* > 0.05); significant *p* values: *, <0.05; **, <0.001; ***, <0.0001. ^‡^ Of the 169 patients, 48 of them had insufficient data to be able to calculate how long they had been under ART.

**Table 2 viruses-15-00027-t002:** HIV-1 variants and drug resistance mutation in ART-treated mother-child pairs.

Mother-Child Pair	HIV-1 Variant		DRM to NRTI		DRM to NNRTI		Major-DRM to PI		DRM to INSTI	
	Mother	Child	Mother	Child	Mother	Child	Mother	Child	Mother	Child
**P1**	**A3**	**A3**	K70R **M184V** K219Q	**M184V**	**K103N** Y181C	**K103N** P225H	none	none	None	none
**P2**	**CRF02_AG**	**CRF02_AG**	**M41L**	**M41L**	none	K103N	none	none	None	none
P3	URF_02BA	CRF02_AG	A62V	M184V	none	K103N	none	none	None	---
P4	**CRF37_cpx**	**CRF37_cpx**	none	none	none	K103N	M46L	none	---	none
P5	CRF02_AG	G	none	M184V	K103N V108I	K103S G190A	none	none	---	---
**P6**	**CRF02_AG**	**CRF02_AG**	M184V	none	**K103N** P225H	**K103N** V108I	none	none	T97A (minor)	---

DRM, drug resistance mutations in 6 mother-child pairs with available *pol* sequences (12PR/12RT/7IN). In bold, subtypes/DRM shared in the mother-child pair; NRTI, nucleoside reverse transcriptase inhibitors; NNRTI, non-NRTI; PI, protease inhibitors; INSTI, integrase strand transfer inhibitors; with dash, *pol* region not available. We considered only major-DRM to PI, NRTI and NNRTI and major + minor to INSTI, according to Stanford v9.0.

## Data Availability

Not applicable.

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
