# Peer review of "High Drug Resistance Levels Compromise the Control of HIV Infection in Pediatric and Adult Populations in Bata, Equatorial Guinea"

_viruses, 2022, doi:10.3390/v15010027_

Round 1

Reviewer 1 Report

The work is devoted to the study of HIV drug resistance in Equatorial Guinea among adults and children, both with and without experience of taking ART. This work opens up new information about the prevalence of HIV DR, as well as genetic variants currently circulating. Unfortunately, such a heterogeneous group is a limitation, because too small cohorts (naive adults, experienced adults, naive children, etc.) do not allow to reliably estimate the prevalence of HIVDR, which the authors talk about.

The authors scrupulously describe the obtained results and draw reasonable conclusions, to which the reviewer has no comments.

At the same time, the reviewer believes that the authors did not carefully study the WHO guidelines on HIVDR. In fact, this is the only, but very important comment to the work.

1. In the Introduction section (lines 87-89) the authors write that WHO recommends studying TDR (transmitted DR) and ADR, citing works published in 2009 and 2015. In fact, currently recommends studying PDR (pretreatment DR) first and not TDR. Authors should review more recent WHO guidelines on HIVDR, such as HIV drug resistance strategy, 2021 update, https://www.who.int/publications/i/item/9789240030565.

2. The authors' poor understanding of the WHO guidelines is evident in chapter 3.4 Transmitted drug resistance mutations by different tools in the Results section. The authors try to compare tools that are used for different purposes. The CPR tool is used for HIVDR surveillance for epidemiological purposes. The CPR tool includes a specific list of epidemiologically significant HIVDR mutations. At the same time, Stanford v9.0 is intended for clinical purposes, identifying HIVDR mutations that are significant in terms of their impact on susceptibility to ARV drugs. Currently, there are many published works that have measured the prevalence of HIVDR using both approaches. However, comparing them is pointless. By the way, according to WHO guidelines, surveillance of the prevalence of TDR (transmitted DR) should be carried out by collecting samples from patients with strictly registered recent infection. The cohorts presented in the work does not correspond to them. Authors should either choose a single analysis algorithm (CPR or Stanford v9.0) or split this chapter into two ones. For example, in the first chapter, talk about the prevalence of HIVDR mutations according to the CPR tool list, and in the second chapter, talk about the prevalence of mutations that cause HIVDR low, intermediate or high level. In this case, the first part will be useful in terms of epidemiological surveillance, the second one will be interesting from a clinical point of view. Following this logic, the phrase in the Discussion section (lines 602-605) should be deleted or heavily reformulated.

Of the not very significant remarks, it should be noted that the authors do not always indicate confidence interval. For example, in the Discussion section, the authors point to a threefold increase in mother to child transmission of HIV (8% versus 2.6%). However, the sample size (2/25) suggests that the confidence interval is huge. Bringing it in this case is absolutely necessary.

Reviewer 2 Report

Abstract

Please provide some more important information in the abstract for example: the rate of delayed diagnosis, prevalence of transmitted and acquired HIV drug resistance (HIVDR) separately, the subtype information in EG and so on.

Introduction

The paragraphs for HIVDR (from p2, line 79 to p3, line 151) are too long. It is advised to simplify and shorten the contents. There is some information not necessary in the introduction or can be moved to the discussion part.

Material and Methods

1. It is advised to use the terms “ART-naïve” or “ART-treated” instead of only “naïve” or “treated” throughout the text, tables, and figures, otherwise please define “naïve” as ART-naïve and treated as ART-treated.

2. Please include the reference for the drug mutation list of INSTI together with the WHO 2009 list for PR and RT.

Tzou et al., Integrase strand transfer inhibitor (INSTI)-resistance mutations for the surveillance of transmitted HIV-1 drug resistance. J Antimicrob Chemother. 2020. 75(1):170-182. doi: 10.1093/jac/dkz417. PMID: 31617907.

3. Please mention what posthoc analysis was applied after the Fisher exact test for 3 groups of comparison for categorical variables.

Results

3.1 Study population

1. Please provide the data for duration of ART, if available.

3.2 ART-failure and treatment delay in HIC-infected and treated population

1. Please describe the definition of “therapeutic failure” in methods section.

2. Please spell out DTG (p8, line 295) as similar to other drugs in the text.

3.3 DRM among 61 naïve and 89 treated patients

1. Please describe the definition of “major” and “minor” drug resistance mutation (DRM) in the methods section.

2. The word ARV-families may not be an appropriate term. ARV-class would be a popular term.

3. The information in the Table 2 would not be important. Please consider moving it to supplement.

4. The data in Figure 4 and the contents in the text (p11, lines 367 to 372) do not match. Please check them carefully once again. Figure 4 would not be necessary if the details of the numbers, percentages, and statistical comparison are properly described in the text.

Table and figures:

Table 1

1. No. (%) appears repeatedly. Please remove it if not necessary.

2. As the total number for each feature is different, thus it is confusing to understand the percentages. It will be reader friendly to add the total number for the features of which the total number is not 237. ex HIV-1 viral load in 169 treated subjects.

Discussion

1. There are many unnecessary and redundant sentences. Please summarize information related to the results of this study only and discuss. Please consider combining paragraphs relating to each other. For example, both the paragraph on delayed diagnosis (lines 500 to 513) and the paragraph on a high rate of mother-to-child transmission (lines 514 to 538) are related to the weak screening/diagnostic strategy/health system in the country. These can be combined and simplified.

2. What is the reason/factor that caused the delay of treatment up to three years in adolescents and adults? Please add a discussion about this finding.

3. Tables 4 and 5 are not necessary to be presented in the discussion section. The current information described in the text is enough for discussion. However, these tables are helpful to understand as a summary. Please consider moving them to supplement.

Round 2

Reviewer 1 Report

Thanks for the changes, I'm happy with them.